# Herpes Simplex Virus Oncolytic Immunovirotherapy: The Blossoming Branch of Multimodal Therapy

**DOI:** 10.3390/ijms21218310

**Published:** 2020-11-05

**Authors:** Laura Menotti, Elisa Avitabile

**Affiliations:** Department of Pharmacy and Biotechnology, University of Bologna, 40126 Bologna, Italy; elisa.avitabile@unibo.it

**Keywords:** herpes simplex virus, oncolytic virus, virotherapy, genetic engineering, virus arming, tropism retargeting, combination therapy, oncolytic immunoviroterapy, immune checkpoint inhibitors

## Abstract

Oncolytic viruses are smart therapeutics against cancer due to their potential to replicate and produce the needed therapeutic dose in the tumor, and to their ability to self-exhaust upon tumor clearance. Oncolytic virotherapy strategies based on the herpes simplex virus are reaching their thirties, and a wide variety of approaches has been envisioned and tested in many different models, and on a range of tumor targets. This huge effort has culminated in the primacy of an oncolytic HSV (oHSV) being the first oncolytic virus to be approved by the FDA and EMA for clinical use, for the treatment of advanced melanoma. The path has just been opened; many more cancer types with poor prognosis await effective and innovative therapies, and oHSVs could provide a promising solution, especially as combination therapies and immunovirotherapies. In this review, we analyze the most recent advances in this field, and try to envision the future ahead of oHSVs.

## 1. Introduction

Herpes simplex virus (HSV) is one of the most studied viruses as an oncolytic virus. From the very beginning, and increasingly with intense research and new developments, it has shown attractive and convenient properties. HSV is a lytic virus, growing at high titers, whose natural history of infection is well known [1]. The genome is a large dsDNA molecule (~152 kbp), and the functions of most of the gene products, encoded by the ~80 open reading frames (ORFs), in virus replication and virus–host interaction have been characterized [1]. Additional cryptic ORFs are being identified by chemical proteomics technology [2]. The entry apparatus, receptor array and entry mechanisms have been largely unraveled [3,4,5]. Many HSV genes are dispensable for growth in cell culture, therefore they can be deleted to accommodate foreign sequences useful for genetic modification, tropism retargeting and virus arming. Furthermore, in the past few decades, the technologies for genetic engineering have been refined, allowing for the construction of the desired recombinants with seamless modifications in bacterial and eukaryotic systems [6,7,8,9,10,11,12,13,14,15]. Last but not least, the specific anti-HSV drug acyclovir represents an excellent safety measure to control the unwanted dissemination of oncolytic HSV (oHSV) infection. In this review, we will cover the past and the state of the art of replication-competent oHSVs. For a comprehensive review of replication defective HSVs and HSV amplicon vectors, see Ref. [16,17,18].

## 2. Attenuated oHSVs

A number of recombinant oncolytic HSVs have been designed and developed in the last years, and have been tested for their oncolytic properties in preclinical models. Many of them entered, and some completed and succeeded in clinical trials (reviewed in [19,20,21,22,23]). The recombinant vectors are often referred to as first-, second- and third-generation oHSVs. However, this classification is not always consistent in the literature. Some authors attribute the oHSV to one or another group according to the number of mutations or genetic modifications (one for first-generation, two for second-generation, or three for third-generation oHSVs) introduced in the genome [24,25]; other authors classify as second-generation oHSVs those carrying mutations and armed with a therapeutic/immunostimulatory transgene [26], and as third-generation the recombinant vectors “containing therapeutic mutations in the genome” [27]. Since it is not possible to merge evenly this nomenclature, we will describe and group oHSVs according to the foremost modification(s) they carry and the underlying oncolytic strategy.

### 2.1. Conditionally Replicating oHSVs with Single or Multiple Mutations

The first oHSVs were generated in the early 1990s. They were mutated or multimutated conditionally replicating recombinants, obtained by the deletion or inactivation of one or more viral genes for nucleotide metabolism, necessary for the virus replication in non-dividing cells (Table 1 and Figure 1) [20]. This engineering strategy restricted the virus replication and final cell lysis to actively replicating cancer cells. Examples of single mutations include the deletion of UL23 (encoding thymidine kinase, TK) in oHSV *dls*ptk [28], or the disruption of UL39 (encoding ICP6, the large subunit of ribonucleotide reductase, RR) in oHSV *hr*R3 [29,30] or of one copy of γ_1_34.5 in NV1020 (R7020) [31]. γ_1_34.5 was recognized as a “neurovirulence gene” encoding a protein able to counteract the innate/intrinsic immunity protein kinase R (PKR) response in non-tumor cells [32], and was one of the cornerstones of oHSV design and development (see below). It soon became clear that UL23 (TK) needed to be maintained in the recombinant genomes for safety reasons, in order to be able to use acyclovir in the case of the unwanted replication of the virus. To reduce the risk of reversion, acquisition of suppressor mutations, or recombination in the host, some of these modifications were combined in multimutated recombinants. Thus, the two copies of γ_1_34.5 (present in the two inverted repeats flanking the UL region of the genome; see Figure 1) were deleted in recombinant oHSV R3616 [32] and 1716 [33,34]; on top of that, UL39 was inactivated in recombinant oHSV G207 by *lacZ* insertion [30]. The 1716 and G207 recombinants proceeded to clinical trials. In a phase Ib trial for recurrent GBM (glioblastoma multiforme), G207 inoculated pre-and post-tumor resection was well tolerated [35]. 1716 has been tested in two phase I clinical trials for high-grade glioma and melanoma [36,37], and more recently in young patients with extracranial solid tumors, showing no toxicity [38].

Interestingly, some of these recombinants were not designed as, or supposed to be, oncolytics from the beginning: constructs with other purposes were at some point assayed for their lytic effect on cancer cells and in animal tumor models (mouse, non-human primates). One prominent case in point is oHSV NV1020 (R7020), an HSV-1/HSV-2 intertypic recombinant. Initially designed as an anti-HSV-2 vaccine candidate, it made it to clinical trials as a candidate oncolytic therapeutic. NV1020 carries the HSV-2 genes from US2 to US8 (for the expression of the gJ, gG, gD and gI glycoproteins as immunogens), plus an α4-*tk* cassette for the expression of TK, in place of most of the inverted repeats joining the UL and US genome regions of HSV-1 (hence just one copy of γ_1_34.5 is deleted) [31]. NV1020 proved safe in mouse and non-human primates, and subsequently proceeded to a phase I clinical trial in patients with colorectal cancer metastatic to the liver and recalcitrant to first-line chemotherapy, and was delivered by infusion in the hepatic artery [39]. The observed safety and tolerability of escalating doses of NV1020 encouraged further phase I/II trials in combination with chemotherapy, whereby a stabilization of the disease course was observed and explained via the action of the resensitization of metastases to chemotherapy [40]. The oHSV HF10 is another example of a serendipitous oncolytic HSV [41,42]; it is a spontaneous HSV-1 mutant lacking the neurovirulent phenotype, derived from the in vitro passaging of an HSV-1 laboratory strain, eventually assayed in trials on head and neck cancers or solid cutaneous tumors [23,43,44,45].

**Table 1 ijms-21-08310-t001:** Conditionally replicating and transcriptionally targeted oHSVs.

oHSV Name (Alternative Name)	Genetic Modification	Diagram in Figure 1, Line	Clinical Trial Identifier (Status)	Ref.
	**Conditionally replicating oHSVs with single or multiple mutations**			
*dls*ptk	deletion of UL23 (encodes TK)	a	–	[28]
*hr*R3	inactivation of UL39 (encodes ICP6, large subunit of RR) by *lacZ* insertion	b	–	[29]
NV1020 (R7020)	deletion of one copy of γ_1_34.5 (encodes ICP34.5 neurovirulence factor, anti-PKR) + HSV-2 US2-US8 genes + α4-*tk*	c	NCT00149396 (C)	[31]
HF10	duplications of UL53, UL54, UL55; deletion of UL56	d	NCT02428036 (C)	[41]
R3616	deletion of two copies of γ_1_34.5 (encodes ICP34.5 neurovirulence factor, anti-PKR)	e	–	[32]
1716	deletion of two copies of γ_1_34.5 (encodes ICP34.5 neurovirulence factor, anti-PKR)	f	NCT00931931 (C)	[33]
G207	deletion of two copies of γ_1_34.5 (encodes ICP34.5 neurovirulence factor, anti-PKR); inactivation of UL39 (encodes ICP6, large subunit of RR) by *lacZ* insertion	g	NCT00028158 (C)	[30]
G47Δ	deletion of two copies of γ_1_34.5 (encodes ICP34.5 neurovirulence factor, anti-PKR); deletion of US12 (encodes ICP47, immune evasion protein); increased expression of US11 (encodes anti-PKR factor)	h	UMIN000015995 (C)	[46]
C134	deletion of two copies of γ_1_34.5 (encodes ICP34.5 neurovirulence factor, anti-PKR); insertion of HCMV IRS1 gene (inhibits antiviral state in the host cell)	i	NCT03657576 (A)	[47]
	**Tumor-specific transcriptionally targeted oHSVs**			
rQNestin (rQNestin34.5v.2)	γ_1_34.5 (encodes ICP34.5 neurovirulence factor, anti-PKR) under control of nestin promoter	j	NCT03152318 (R)	[48]
NG34	GADD34 (human counterpart of γ_1_34.5) under control of nestin promoter	k	–	[49]

TK: thymidine kinase; PKR: protein kinase R; RR: ribonucleotide reductase; NCT: trials registered at ClinicalTrials.gov; UMIN: trials registered in University hospital Medical Information Network (Japan). (A): active, not recruiting; (C): completed; (R): recruiting.

The demonstration that attenuated replicating oHSVs could work prompted data-driven approaches for the development of multimutated recombinants more efficacious in tumor killing. A noteworthy modification was the deletion of US12, encoding ICP47, originally identified as a compensatory mutation in Δγ_1_34.5 viruses [50,51]. The effects of this engineering were twofold. First, the attenuation of Δγ_1_34.5 recombinants was reduced, and the viruses displayed increased virus replication and improved lysis of tumor cells [46]: this effect depends on the new kinetic of expression of US11 (normally a late gene, now an immediate-early gene under control of US12 promoter), which precludes the phosphorylation of eIF-2α, curtailing the PKR pathway of shutoff of protein synthesis. Second, ICP47 is an immune evasion protein blocking antigen presentation in infected cells: thus, tumor cells infected with a ΔUS12 virus expose more efficiently viral and tumor antigens on the plasma membrane and are more immunostimulatory than cells infected with a wild type (wt) virus. These astounding combined properties, deriving from the deletion of a single gene, were widely exploited for the educated construction of triply-mutated oHSVs, like oHSV G47Δ and oHSV T-01 [25,46,52,53,54]. Notably, US11 expression by G47Δ recombinant rescued viral replication in glioblastoma stem-like cells (GSCs, more related to neural stem cells, NSCs), where replication of Δγ_1_34.5 recombinants, like G207, was restricted [55,56]. Indeed both G47Δ and G207 were able to replicate in matched serum-cultured GBM cells (ScGCs, more differentiated than GSCs) [56], indicating that US11 expression by G47Δ has a pivotal role for virotherapy efficacy in less differentiated glioblastoma cells. G47Δ has been tested in Japan in five phase I or II clinical trials (completed or recruiting) for recurrent glioblastoma, prostate cancer, olfactory neuroblastoma and pleural mesothelioma, and is currently in a fast-track for drug approval in Japan [57].

A key point in oHSV design was the improvement of replication of attenuated oHSVs, necessary to reach adequate titers both in virus preparations and in the target tumor to maximize therapeutic efficacy. With safety in mind, the deletion of HSV γ_1_34.5 was restored or circumvented in two ways. First, tumor-specific transcriptional targeting was used to boost virus replication in tumor cells, ideally without causing toxicity to healthy cells, by placing the γ_1_34.5 gene expression under control of a promoter expressed predominantly in tumor cells. An example is the nestin promoter/enhancer, expressed in glioma cells, engineered to control γ_1_34.5 expression in the oHSV rQNestin34.5 [58]. A phase I clinical trial with rQNestin (rQNestin34.5v.2), the investigational new drug (IND) version of the oHSV, is at present recruiting (NCT03152318) [48]. Second, HSV γ_1_34.5 was substituted with human or viral anti-PKR genes. The oHSV NG34 is an improved version of rQNestin34.5 recombinant, with further reduced toxicity, bearing GADD34, the human counterpart of γ_1_34.5, under the nestin promoter/enhancer [49]. A second strategy was to exploit human cytomegalovirus genes IRS1 and TRS1, functionally analogous to γ_1_34.5 and involved in cellular antiviral state inhibition, to engineer HSV-HCMV chimeric viruses exhibiting enhanced replication, without the full rescue of the neurovirulent phenotype typical of wt HSV [47]. In particular the oHSV C134, carrying the IRS1 gene of HCMV, was the most promising recombinant, able to prolong survival in mouse models of GBM, without sign of neurovirulence [59]. The clinical trial that will test C134 on recurrent GBM is now recruiting (NCT03657576).

### 2.2. Armed oHSVs

The oncolytic strategy of the multimutated oHSV relied on the selective lysis of tumor cells following tumor cell-specific virus replication, and on the block of replication in normal non-cancer cells. However, this approach, taken for the sake of safety, caused in most of the multimutated oHSVs an attenuation (i.e., reduced replication) in tumor cells, as compared to wt HSV. To circumvent this drawback, an innovative approach was to “arm” the recombinant backbones carrying the attenuating mutations with transgenes in order to restore oncolytic efficacy notwithstanding reduced replication [26,60]. A number of different strategies were explored (Table 2 and Figure 2): the insertion of sequences encoding heterologous fusogenic glycoproteins to induce syncytia formation [61], e.g., GALV envelope fusogenic membrane glycoprotein [62]; the expression of suicide genes, e.g., rat cytochrome P450 2B1 (CYP2B1) activating the prodrug cyclophosphamide [63], or yeast cytosine deaminase, converting the prodrug 5-fluorocytosine (5-FC) to cytotoxic 5-fluorouracil, as standalone yCD [64], or as Fcy::Fur, in fusion with UPRT (uracil phosphoribosyltransferase) [65]; the expression of anti-angiogenic, proapoptotic or spread-enhancing proteins [60,66,67,68].

The most important and effective arming was achieved with the engineering of recombinant genomes expressing immunomodulatory molecules. Arming with cytokines is deemed a “*cis*-combination therapy” aimed at immunologically stimulating the tumor microenvironment [75]. In fact, following the observation that the in vivo administration of (multi)mutated oHSVs could induce local and systemic antitumor immunity and elevate specific CTL responses [76], it was expected that the administration of oHSVs expressing different immunomodulatory molecules would enhance the effects of the immune system and cooperate with the intrinsic lytic activity of the virus, overall potentiating the oncolytic efficacy. Thus, the engineering of murine IL-4 in place of both copies of γ_1_34.5 improved the in vivo efficacy of the recombinant oHSV R8306 in brain tumor models in mice [69]. Major developments came from the expression of IL-12 by oHSVs M002 and NV1042, which were effective in murine and primate models of brain tumors [70,77,78,79,80], squamous cell carcinoma [72], colorectal tumors [81], or spontaneous prostate tumors [82,83]. Mice survival was prolonged, and the tumors were infiltrated by NK, CD4+, CD8+ T cells and macrophages. The oHSV M032, the human-IL-12-expressing version of M002, has been tested for safety and stability in non-human primates by intracerebral administration [71,84]. At present, a phase I clinical trial with M032 is recruiting patients with recurrent/progressive GBM, anaplastic astrocytoma, or gliosarcoma, with an estimated study completion date of September 2023 (NCT02062827), and phase I trials on animals have been designed [85]. A multipronged engineering strategy with immunostimulatory genes has been assayed too—the combined administration of three oHSVs armed with different cytokines (IL-12, IL-18) or soluble CD80 (B7-1) showed a greater oncolytic effect relative to the administration of the same cumulative dose of just a single type of armed oHSV [86].

The engineering with granulocyte macrophage colony-stimulating factor (GM-CSF) deserves a special insight (see Section 2.3), because it had different outcomes in different HSV backbones and experimental settings.

### 2.3. Talimogene Laherparepvec (T-VEC)

Besides the promising IL-12, GM-CSF was also evaluated as an “arming cytokine” for engineering immunostimulatory oHSVs. GM-CSF is a myelopoietic growth factor with pleiotropic effects, such as the induction of immature myeloid cell differentiation, and mature myeloid cell (polymorphonuclear neutrophils, monocytes/macrophage and dendritic cells) recruitment and activation. According to today’s literature, GM-CSF’s complex mechanisms of action are still not fully elucidated, and may end—contradictorily—in immunostimulation or immunosuppression in different physiological or pathological situations (infection, inflammation, cancer) [87]. Retrospectively, it is not surprising that early studies on oHSVs engineered with GM-CSF yielded contradictory results. The oHSV NV1034 expressing murine GM-GSF (from NV1020, with an HSV-1/HSV-2 intertypic recombinant genetic background) did not display an oncolytic enhancement in models of squamous cell carcinoma and prostate cancer, and was clearly less efficacious than the murine IL-12-expressing counterpart NV1042 [72,88]. On the contrary, defective (non-replicating, non-lytic) HSV vectors, engineered to express murine GM-CSF as in vivo cytokine gene transfer vehicles at tumor sites, showed efficacy for active cancer in situ immunotherapy and as systemic tumor vaccines, eliminating the toxicity associated with the systemic administration of recombinant cytokines [89,90].

The best success of GM-CSF engineering was obtained with the oHSV initially designated JS1/ICP34.5-/ICP47-/GM-CSF, carrying murine GM-CSF in place of the two copies of γ_1_34.5 [73]. Of note, this recombinant was purposely designed to achieve enhanced anti-tumor potency with a specific strategy—first, it was obtained starting from a clinical isolate (JS-1) displaying enhanced cell killing of an array of human tumor cell lines, as compared to laboratory strains (e.g., HSV-1 strain 17); second, its genome was also modified with the deletion of US12 (ICP47), as described above for G47Δ (see Section 2.1) to enhance replication, immunostimulation and tumor lysis. Overall, these modifications worked synergistically, and the recombinant oHSV had a prominent immunostimulatory profile, and worked as a “cancer vaccine” protecting against distant tumors and metastases, or tumor rechallenge in a mouse subcutaneous model of lymphoma [73]. The oHSV version carrying human GM-CSF engineered for clinical trials was named OncoVEX^GM-CSF^, and finally talimogene laherparepvec (T-VEC, trade name Imlygic). T-VEC has undergone phase I and II clinical trials in different solid tumors and disease conditions, alone as a monotherapy [74,91,92,93] or as a combination therapy with cisplatin and radiation [94]. A systemic anti-tumor immunity was elicited, and determined the regression of non-injected tumors. In a phase III clinical trial (OPTiM), T-VEC showed efficacy as a monotherapy against advanced melanoma, compared to a standard GM-CSF treatment [95,96,97,98,99,100,101]. The trial assessed T-VEC biodistribution, and shedding as well [102]. Following these clinical results, T-VEC was the first “oncolytic immunotherapy” (not just “oncolytic virus” or “oncolytic virotherapy”) to be approved by the FDA (October 2015) and by the EMA (December 2015) [103,104]. As a follow-up, a clinical trial aimed at monitoring the survival and long-term safety of patients who received at least one dose of T-VEC, regardless of the type of tumor, will recruit about 300 subjects, and is planned to be completed in 2023 (NCT02173171).

## 3. Tropism Retargeted, Unattenuated, oHSVs

In some instances, multimutated and/or armed recombinant oHSVs achieved cancer specificity at the cost of attenuation, with a potency in cancer cell killing well below the wt viruses they were derived from. Attenuation was the essential condition to step into clinical trials and for allowing ethics committees and regulatory agencies to start trusting viruses as potential, and especially safe, therapeutics [103]. Tropism retargeting is an alternative strategy that has been envisioned, and applied, to maintain the full lytic potential of HSV, while conferring cancer specificity, and therefore safety. This approach exploits the expression of cancer-specific antigens at the plasma membrane of selected tumors, combined with a deep knowledge of virus natural receptor recognition and binding, and entry mechanisms. The latter allows the design of specific modifications of the viral components involved in entry in order to retarget recombinant virus entry to cancer cells only, via the tumor-specific receptors. For HSV, this has been a promising approach for a number of reasons [105]. First, HSV exploits the viral glycoproteins inserted in the viral envelope to achieve attachment and entry. Of note, viral glycoproteins are flexible and motile in the viral lipid bilayer [106]; in the perspective of engineering, this is an advantage as compared to the rigid capsid proteins of non-enveloped viruses, which impose more strict steric and spatial constraints. Second, both the interplay of viral and cellular components at the plasma membrane in initiating entry mechanisms, and the subsequent downstream signaling or intracellular events, have been investigated in depth at the biological, cellular, biochemical, molecular and structural levels. HSV has a set of four envelope glycoproteins essential for virus entry: gD (involved in specific receptor recognition and binding, and the triggering of the fusogenic signal to downstream effectors), gB (a class III fusogenic glycoprotein which executes fusion), and the heterodimer gH/gL (partnering gB to execute fusion, and lacking any similarity with any other fusion protein) [3,107,108,109,110]. Recently, insights into additional gB domains spatially distant from the fusion loops have come from structural studies on the related alpha-herpesvirus varicella zoster virus (VZV) [111]. These glycoproteins exploit a wide array of natural receptors for target cell recognition and fusion execution. Thus HVEM/HveA, Nectin-1/HveC and 3-OS HS are receptors for gD, PILRα, MAG and NMHC-IIA and B are bound by gB, and gH/gL interact with integrins α_V_β_3_, α_V_β_6_ and α_V_β_8_ [5,109,112,113,114,115]. This multipartite system allows the virus to penetrate via two different pathways—by fusion at the plasma membrane or by endocytosis [4,116,117,118].

From this strong, still growing, background, mainly two different strategies have been envisioned to retarget HSV tropism. Here, we just mention the approach exploiting bispecific adapters (bridging proteins) bound to viral particles to retarget HSV to heterologous receptors; the pros and cons of this methodology are discussed in depth elsewhere [119]. In this review, we will principally focus on the HSV tropism retargeting strategies that involved the engineering of recombinants carrying genetic modifications, transmissible to the viral progeny, and on how the recombinants were progressively improved to attain “full retargeting”, i.e., retargeting to tumor receptors and detargeting from natural receptors.

The most successful retargeting strategies relied on the modification of glycoproteins essential in HSV entry, namely gD, and the trio gB-gH/gL (Table 3, Figure 3 and Figure 4).

The first breakthrough was the evidence of the possibility of redirecting HSV-1 to a heterologous receptor, namely IL-13Rα2, expressed in malignant glioma, by inserting IL-13 in gD at amino acid 24 [120]. In this first engineering approach, to facilitate the binding to the target receptor, the virus was also modified in gC, deleting the HS-binding moieties (normally involved in non-specific attachment to cells), and inserting a second copy of the targeting ligand IL-13. The natural attachment to HS on the cell surface was further disabled by deleting the polylysine tract domain of gB. The recombinant virus R5111 was able to infect cells expressing solely IL-13Rα2 as receptor, however it was still able to enter cells via HVEM/HveA or Nectin-1. After this proof-of-principle, other receptors, e.g., uPAR, were targeted by a similar genetic engineering strategy in oHSV R5181 gD [121], demonstrating that HSV could be retargeted to heterologous receptors belonging to different molecular families through different types of ligands, like a cytokine (IL-13) or part of a protease (uPA). This hinted that gD was quite flexible, and tolerated insertions without losing its pro-fusogenic activity.

A key issue was to achieve detargeting, i.e., the abrogation of HSV natural tropism, in order to derive recombinant vectors specific for the tumor cells, which would spare healthy cells. This was a tough challenge, since HSV receptors for gD, especially Nectin-1, are ubiquitously expressed in human and rodent cells and tissues. The IL-13Rα2 retargeted recombinant vector was improved by the deletion of the entire gD N-terminus (N-ter) amino acid residues 1-32, mapped as the HVEM/HveA binding site, and by the point mutation V34S, which disrupted the interaction with Nectin-1 (oHSV R5141) [122]. The latter substitution worked for the IL-13 engineering but was not universal [124].

The research on retargeting to HER2, a receptor overexpressed in breast and ovarian cancers, led to a turning point via two stepwise pieces of evidence. First, the insertion of a single chain antibody (scFv) to HER2, corresponding to trastuzumab/Herceptin, was tolerated by gD. This insertion at the N-ter between amino acids 24 and 25 conferred retargeting to the heterologous cancer-specific receptor, and abrogated entry via HVEM/HveA, but left the interaction with Nectin-1 unhampered (oHSV R-LM11) [123]. Second, the insertion of the scFv directed to HER2 in place of two large deletions, at the flexible N-ter (Δ6-38, in oHSV R-LM113) or in the gD core (Δ61-218, in oHSV R-LM249), could also detarget the recombinant virus from both HVEM/HveA and Nectin-1 [124,125]. The recombinant oHSVs were not attenuated and replicated to high titers, and they were safe and effective in subcutaneous or intraperitoneal mouse models of primary or metastatic ovarian carcinoma [125,132], or in infiltrative glioma models transplanted intracranially into immunodeficient or immunocompetent mice [133,134,135]. The modification (deletion+insertion) at the N-ter of gD constitutes a platform for the generation of fully retargeted oHSVs, as was demonstrated by the engineering of different scFvs directed to EGFR (oHSV R-611), PSMA (oHSV R-593) and EGFRvIII, a glioma-specific variant of EGFR (oHSV R-613) [127]. In addition, the oHSVs were further equipped and armed. Reporter genes (EGFP, Gaussia Luciferase) were added without harming the in vitro oncolytic activity of recombinant vectors [127], and indeed arming with the immunostimulatory gene encoding IL-12 enhanced the in vivo efficacy via a robust improvement of the tumor immunological microenvironment (oHSV R-115) [14,136,137]. The oHSVs retargeted to EGFR and EGFRvIII (oHSV KNE), or CEA (oHSV KNC), obtained by a similar strategy of scFv engineering into gD, were further developed focusing on the potentiation of virus entry and spread by substitutions in gB and matrix metalloproteinase-9 arming [126,138,139], and on preventing off-target replication by means of miRNA response sequences [140,141]. In this regard, recently a combined strategy of tropism retargeting and tumor-restricted replication was applied by placing the essential immediate early α4 gene of a fully virulent HER2-retargeted oHSV under the control of the Survivin/*BIRC5* promoter, which is highly transcribed in cell cycle phase G2 [142].

In retrospect, the Nectin-1 detargeting for the engineering at the N-ter was attributed to the deletion of gD aa residue 38, as confirmed by the gD–Nectin-1 co-crystal structure [143]. This insight allowed more circumscribed deletions and refined insertions. Thus, gD can accommodate two inserts in the same molecule, and can be retargeted to two different receptors simultaneously (oHSVs R-87, R-89, R-97, R-99) [128]. Moreover, the retargeting options are not limited to gD, but can be expanded to other glycoproteins essential for entry, namely gB (oHSV R-909) and gH (oHSVs R-VG809 and R-213) (Table 3 and Figure 4) [129,130,131]. gH and gB tolerate the insertion of a peptide or of a scFv as heterologous ligand, and work in retargeting the oHSV tropism to a tumor receptor. In particular, with the modification of gB with an scFv directed to HER2 (in R-909), the multipartite entry apparatus of HSV was converted to monopartite apparatus, whereby the functions of receptor binding and fusion execution were combined in the sole gB, which still involved a mutated form of gD (unable to bind its natural receptors, thus non-activable) in a proposed “structural” role in the entry quartet complex gD, gB and gH/L [131]. Recent in vitro work on retargeted gD carrying point mutations has revealed the possibility of engineering mutant gDs able to escape virus-neutralizing antibodies. While this approach would allow, in perspective, a more efficient systemic delivery, at present it has the disadvantage of a reduced incorporation of the mutant gD in the virion envelope, which could in practice undermine the efficiency of entry into target cancer cells [144].

## 4. oHSV Delivery

oHSV route of administration and delivery to the target tumor site has been taken into account as a key issue from the initial design of recombinant viruses. Intratumoral delivery is certainly the most efficient route, as it avoids the loss of therapeutic virus and limits the amount needed per treatment. In the case of T-VEC, it is optimal because it can induce an inflammatory microenvironment in the tumor, prompting systemic immunity. However, intratumoral delivery may not be possible for some body sites, and may not be efficacious for treating disseminated metastases. On the other hand, intravenous delivery has theoretically the potential to reach most body organs, but it must face two issues, i.e., the high HSV seroprevalence in the human population (80–90%) and the amount of inoculum to be administered for the oHSV to reach the target tumor cells at a sufficient quantity. Preclinical mouse models were intensively used to untangle all these concerns. As for the interference of the immune system, encouraging results have come from studies now dating back almost a couple of decades, wherein it was shown that systemic (intravenous) delivery is feasible and there are no significant effects due to seroconversion [73,83,145]. However, for the translation to patients, manufacturing and characterization challenges exist [146]. They are linked to the virus particle large size and composition, with the absolute requirement of preserving the envelope, and are related to the structure of the genome, which is arranged in four possible isomers. oHSV preparations with high purity and titer are obtained from up to 100 L of supernatant of GMP-certified producer cells (e.g., Vero, an African Green Monkey cell line) infected with a characterized “seed virus”. Progeny virions are harvested by ultracentrifugation or tangential flow filtration, followed by size exclusion chromatography and ion-exchange chromatography. Contaminating DNA is removed by Benzonase treatment. Sterile filtration precedes the filling of the final product [146]. Viral titers obtained are in the range of 1 to 5 × 10^9^ PFU/mL [71,146]. Therefore, it must be considered that the amount of virus that should be administered to patients to circumvent the dilution effect in the bloodstream, or clearance by the liver or the mononuclear phagocyte system, is at present well above the manufacturing possibilities, or collides with safety concerns about the needed oHSV concentration in blood [103]. For this, “shielding” approaches must be envisioned [147]. In this regard, mesenchymal stromal cells (MSCs) have proven successful as carrier cells in a model of ovarian cancer metastatic to the lung [148], and interestingly, the incorporation of a CD47 “don’t eat me” signal molecule in the viral envelope of an HSV-2-based oHSV was effective in promoting carrier-free delivery to, and persistence at, tumor sites [149]. Models of peritoneal or meningeal metastases showed the advantages of systemic/loco-regional intraperitoneal or intra-cerebrospinal fluid delivery of oHSVs [132,150]; in particular, the intraventricular administration of oHSV displayed an advantage over chemotherapy because of its slower clearance from the cerebrospinal fluid with an improved therapeutic outcome [150]. Alongside these results, in a clinical perspective, the experience gained with T-VEC [74,95,151] indicates that at present loco-regional delivery is an attained and concrete therapeutic option, which will provide systemic efficacy by the release of tumor antigens and the subsequent recruitment and cooperation of the immune system [103].

## 5. oHSV Combination Therapies and Immunotherapies

Taken together, these findings pave the way to the bottom-up generation of fully virulent, fully retargeted, armed, safe and effective oHSVs, with the desired features for the specific treatment of tumors or metastases recalcitrant to standard therapies. However, the occurrence of therapeutic resistance observed in preclinical models, or some limited results observed in the clinics, preclude the use in the long run of oHSV as a monotherapy [27,152]. The emerging scenario envisages oHSVs as a partner in multimodal therapies, in combination with treatments whose properties (efficacy and side effects) are fairly well known; these include standard chemo- and radio-therapy, and more recent and innovative immuno-therapies. The advantage of placing side by side oHSV and standard therapies is that an enhanced efficacy can be readily ascribed to the combination of the two, and some side effects can be readily recognized. In addition, the possibility of making the oHSV an armed carrier of the therapy itself, has in principle the advantage of confining the site of drug action, limiting systemic side effects.

To date, a number of investigations into the multimodal therapy of oHSV associated with chemotherapy have been conducted [75,152]. In most instances, the immune system is the major effector in tumor clearance, after the oHSV-mediated release of tumor antigens. Indeed, the outcome of experimental virotherapy depends on the level of infiltration of both tumor- and virus- antigen specific cytotoxic T cells [153]. Thus, the understanding of the tumor micro-environment and its regulation by locally or systemically administered drugs/therapeutics, or transgenes expressed in situ from the oHSV itself, is of fundamental importance. A case in point is rRp450, an armed oHSV engineered with rat CYP2B1, a prodrug-converting enzyme for the immunosuppressive drug cyclophosphamide [154]. More recent studies show and confirm that the combinatorial effect may provide a reciprocal advantage to chemotherapy, or oHSV alone [75]. In fact, some tumors may be, or may become, resistant to oHSV due to the pro-inflammatory response of tumor-associated macrophages (via a constitutively activated MEK or STAT signaling) in the tumor milieu, which blunts oHSV replication. The pharmacological blockade of these intracellular pathways in tumor-associated macrophages, and the overall modulation of the host inflammatory response to the virus, can restore oHSV replication in tumor cells and the consequent oHSV-related CD8+ T cell activation, responsible for the antitumor effect observed in animal models [155,156,157]. In turn, the alteration of the tumor microenvironment, following oHSV replication, can enhance the chemotherapeutic drug’s diffusion to (and efficacy against) the tumor.

The combination with radiotherapy has been explored since the development of the first oHSVs [158]. Radiation increases tumor antigen release and cooperates with oHSV. A completed phase I clinical trial on G207 oHSV showed its safety and efficacy against glioblastoma [159]. More trials of oHSVs in combination with radiation on pediatric brain tumors (G207, phase I) or other solid tumors (T-VEC, phase II) are presently active or recruiting [75,160].

An expanding field is the combination with immune-checkpoint inhibitors (ICIs), like anti-PD-1, anti-PD-L1 or anti CTLA-4 antibodies. CTLA-4 and PD-1 are immune checkpoint (IC) proteins which regulate T cell activity by two independent inhibitory pathways: CTLA-4 blocks early T cell activation and stimulates regulatory T cells (Tregs), while PD-1 blocks killing by T effector cells [161]. In tumor pathophysiology, the binding between PD-1 on T cells and PD-L1 on cancer cells, or between CTLA-4 on T cells and B7 on tumor cells, causes the escape/evasion of the tumor from the immune response and supports the growth of malignant cells. The blockade of the PD-1/PD-L1 and/or of the CTLA-4/B7 axes by antibodies reinstates T cell function and anti-tumor cytotoxic activity in the tumor microenvironment. The acquired resistance to ICIs and/or toxicities caused by ICI therapy has been reported. Therefore, the ICI therapeutic intervention can benefit from combination with other regimens. Oncolytic viruses are prominent candidate partners for ICI therapies as viral infection and oncolysis cause tumor-antigen release, induce anti-tumor immune responses, alter immunologically the tumor microenvironment, and can increase the efficiency of checkpoint inhibition, which are known to work better in immunologically active sites. Thus, a so-called “cold” (immunosuppressive, with few immune effector cells) tumor microenvironment can be primed by viral infection and turned into a “hot” (immunogenic, highly infiltrated by immune effector cells) tumor microenvironment [162,163,164,165,166,167,168]).

The ICIs+oHSV combination strategy has been envisaged, and received strong support in the last decade [169,170,171,172,173]. Thus, T-VEC (see Section 2.3) has been assayed in clinical trials for advanced melanoma in combination with ipilimumab (directed to CTLA-4), with an objective response rate of 39%, [174,175], or in combination with pembrolizumab (anti-PD-1), with a complete response rate of 33% [98,176]. RP1, a GALV fusogenic protein, and GM-CSF armed oHSV are presently being assayed with nivolumab (anti-PD-1) in solid tumors (NCT03767348) [177]. A phase I trial on canine patients with IL-12-expressing M032 oHSV in combination with a checkpoint inhibitor has been designed [85]. In a preclinical setting of a more “cold”, non-immunogenic tumor, such as GBM, G47Δ showed the best efficacy when in “triple combination”, i.e., administered simultaneously with both anti-PD-1 and anti-CTLA-4 antibodies, with up to 89% long-term survivors, the establishment of immunological memory, and the lack of recurrence of the tumor [100,178,179]. As a further development, oHSV could be armed with ICIs for the in situ targeted delivery of the combination therapy. In this regard, the NG34 oHSV (see Section 2.1) was engineered to express an scFv directed to PD-1, and conferred prolonged survival in syngeneic immunocompetent mouse models of GBM and the establishment of an anti-tumor memory response [180]. These preclinical data and trials indeed indicate that tuning the delicate balance and interplay between ICIs and oHSVs can work as a personalized medicine strategy of cancer vaccination to patient-specific, autologous tumor antigens.

## 6. Conclusions

Big advances have been made in the development of HSV-based oncolytic viruses and the related therapeutic strategies, making HSV a forerunner in the field. Many other oncolytic viruses belonging to different virus families (adenovirus, measles virus, reovirus, vaccinia virus, parvovirus) are already at the clinical trial stage, and accompany oHSVs in their undertaking towards innovative, effective and personalized antitumor strategies [181,182]. The mentioned different virus families have diversified properties, but share similar challenges, notably their intricate relationship with the immune system of the host [162,170,183]. A definite challenge for oHSVs is the optimization of systemic delivery. Still, the numerous advantages and the versatility of oHSVs, along with the multifaceted engineering possibilities and the promising opportunities of combination with standard and innovative therapies, will foster upcoming exciting developments towards patient-tailored anti-cancer therapies and vaccination.

## Figures and Tables

**Figure 1 ijms-21-08310-f001:**
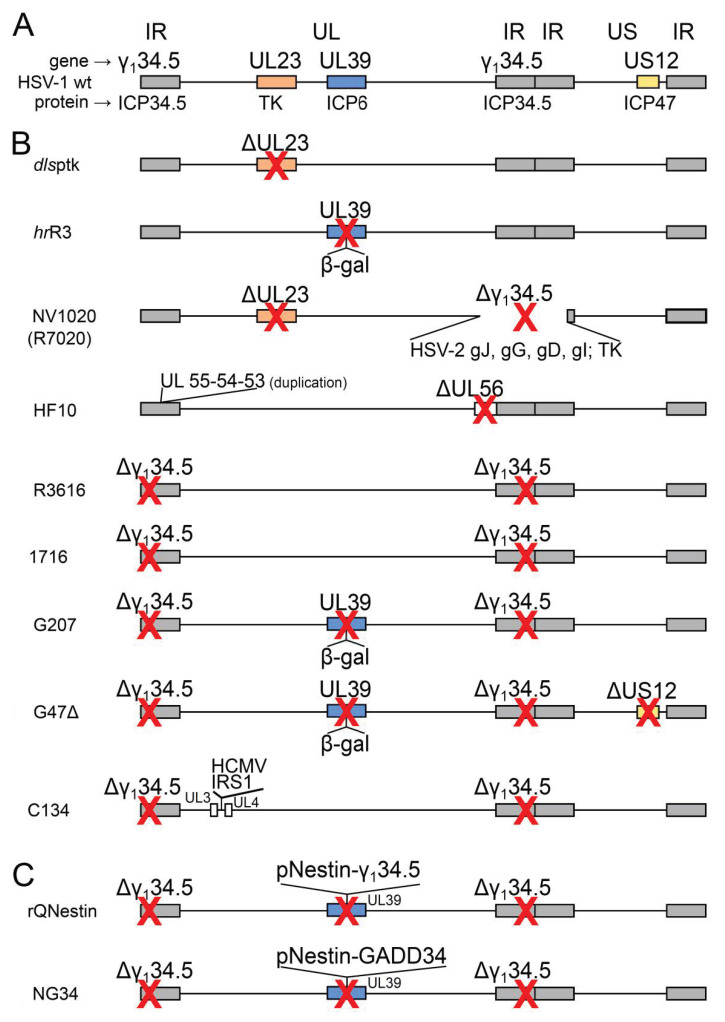
Schematic diagram of wt and recombinant HSV genomes. Genes relevant for tumor-specificity are shown as colored boxes and their names are indicated above the genome line; the gene product names are indicated below the genome line. (**A**) wt HSV: IR, inverted repeats (grey boxes); UL: unique long; US: unique short. (**B**,**C**) Diagrams of the oHSVs described in Table 1: (**B**) oHSVs with single and multiple mutations, (**C**) transcriptionally targeted oHSVs. β-gal: β-galactosidase; TK: thymidine kinase. Red crosses indicate the inactivation of a gene, either by deletion (Δ) or by insertion.

**Figure 2 ijms-21-08310-f002:**
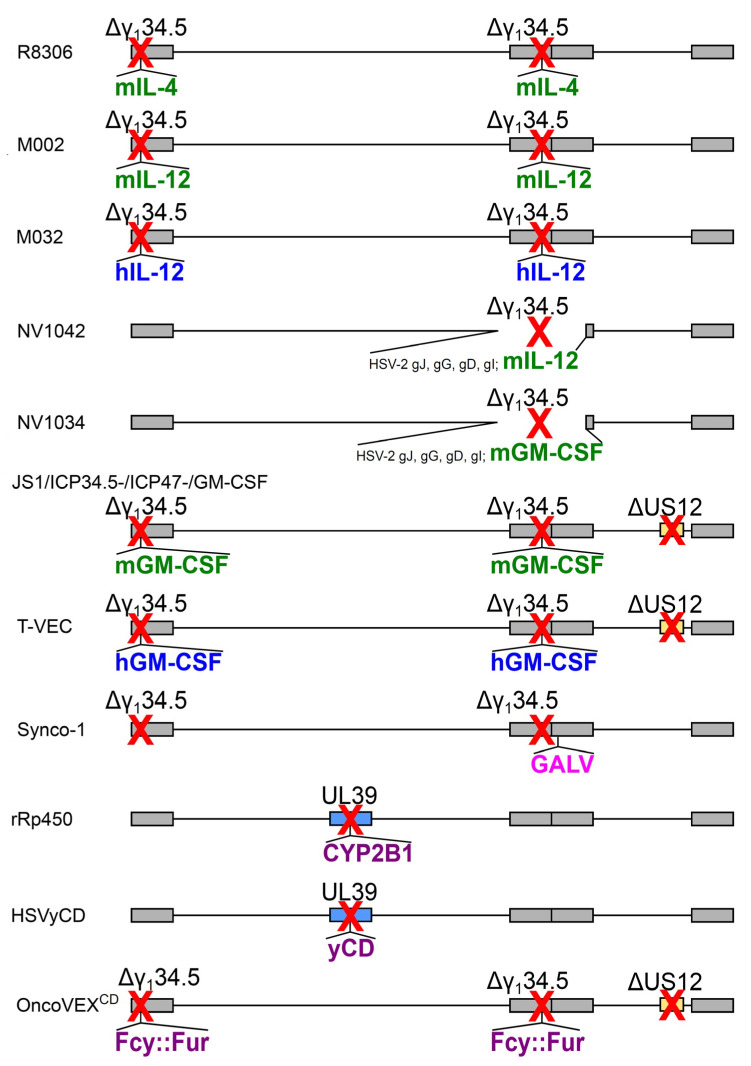
Schematic diagram of example armed oHSVs genomes described in Table 2. Murine cytokines are in green font, human cytokines in blue, heterologous fusogenic proteins are in pink, prodrug converting enzymes in purple. Red crosses indicate the inactivation of a gene, either by deletion (Δ) or by insertion of the heterologous arming gene.

**Figure 3 ijms-21-08310-f003:**
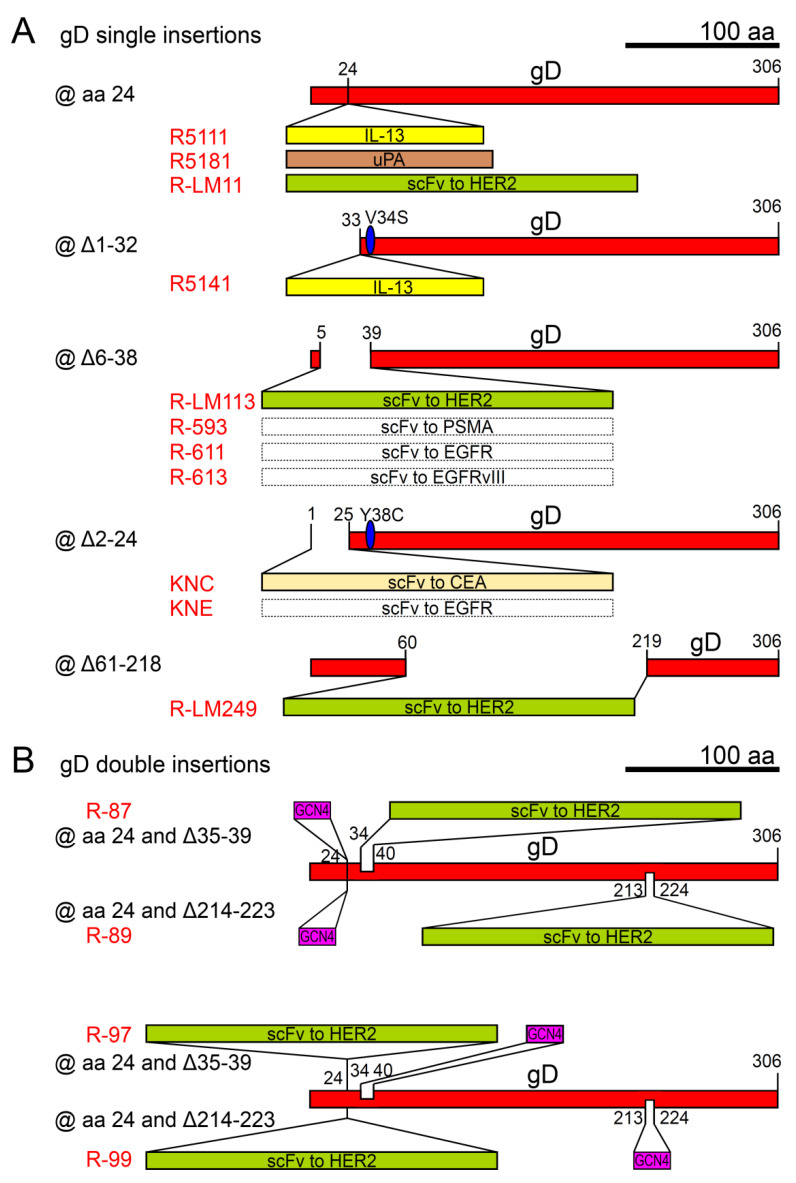
Schematic diagram of the ectodomain of HSV gD engineered for tropism retargeting, described in Table 3. (**A**) gD with single insertions, (**B**) gD with double insertions. The names of the oHSV recombinants are in red. Numbers indicate amino acid (aa) residues. The boxes depicting gD or insert lengths are drawn to scale. The blue oval indicates a mutated residue. Scale bar: 100 aa. Δ: deletion.

**Figure 4 ijms-21-08310-f004:**
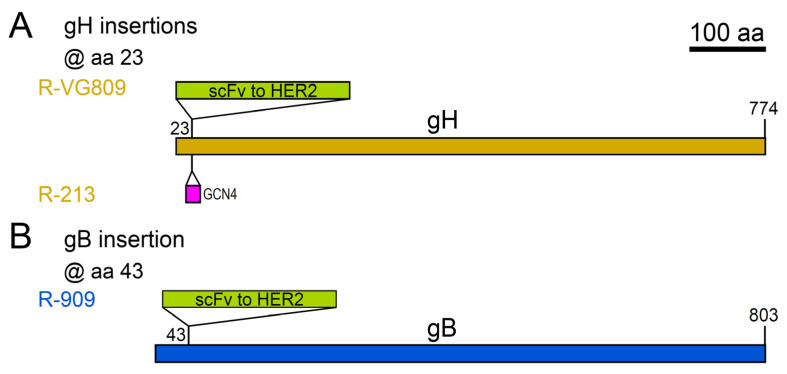
Schematic diagram of the ectodomain of the HSV glycoproteins gH (**A**) and gB (**B**) engineered for tropism retargeting, as described in Table 3. Both gH and gB carry single insertions. The names of the oHSV recombinants are in ochre yellow for gH engineering and blue for gB engineering. Numbers indicate amino acid (aa) residues. The boxes depicting glycoproteins or insert lengths are drawn to scale. Scale bar: 100 aa.

**Table 2 ijms-21-08310-t002:** Characteristics of example armed oHSVs.

oHSV Name	Expressed Transgene @ Viral Locus	Parental Virus	Diagram in Figure 2, Line	Clinical Trial Identifier (Status)	Ref.
R8306	murine IL-4 @ γ_1_34.5 loci	HSV-1(F), Δ 2 copies of γ_1_34.5	a	–	[69]
M002	murine IL-12 @ γ_1_34.5 loci	HSV-1(F), Δ 2 copies of γ_1_34.5	b	–	[70]
M032	human IL-12 @ γ_1_34.5 loci	HSV-1(F), Δ 2 copies of γ_1_34.5	c	NCT02062827 (R)	[71]
NV1042	murine IL-12 @ γ_1_34.5 locus	NV1020 (Δ 1 copy of γ_1_34.5)	d	–	[72]
NV1034	murine GM-CSF @ γ_1_34.5 locus	NV1020 (Δ 1 copy of γ_1_34.5)	e	–	[72]
JS1/ICP34.5-/ICP47-/GM-CSF	murine GM-CSF @ γ_1_34.5 loci	JS-1 ^1^, Δ 2 copies of γ_1_34.5, ΔUS12	f	–	[73]
OncoVEX^GM-CSF^, T-VEC, talimogene laherparepvec	human GM-CSF @ γ_1_34.5 loci	JS-1 ^1^, Δ 2 copies of γ_1_34.5, ΔUS12	g	NCT00769704 (C)	[74]
Synco-1	GALV fusogenic protein @ packaging signal	HSV-1, Δ 2 copies of γ_1_34.5	h	–	[62]
rRp450	rat CYP2B1 @ UL39	HSV-1 (KOS) + inactivated UL39	i	NCT01071941 (R)	[63]
HSVyCD	yCD @ UL39	HSV-1 (KOS) + inactivated UL39	j	–	[64]
OncoVEX^CD^	Fcy::Fur fusion @ γ_1_34.5 locus	JS-1 ^1^, Δ 2 copies of γ_1_34.5, ΔUS12	k	–	[65]

^1^ HSV clinical isolate. Δ: deletion. NCT: trials registered at ClinicalTrials.gov. (C): completed; (R): recruiting.

**Table 3 ijms-21-08310-t003:** Characteristics of example tropism retargeted oHSVs sorted by engineered viral glycoprotein.

Retargeting Ligand(s) @ Viral Glycoprotein	Target Heterologous Receptor	oHSV Name	Parental Strain	Ref.
**@ gD**				
IL-13 @ gD aa 24	IL-13Rα2	R5111	HSV-1(F)	[120]
uPA @gD aa 24	uPAR	R5181	HSV-1(F)	[121]
IL-13 @ gD Δ1-32	IL-13Rα2	R5141	HSV-1(F)+gDV34S	[122]
scFv to HER2 @ gD aa24	HER2	R-LM11	HSV-1(F)BAC+*lacZ*	[123]
scFv to HER2 @ gDΔ6-38	HER2	R-LM113 ^1^	HSV-1(F)BAC+EGFP	[124]
scFv to HER2 @ gDΔ61-218	HER2	R-LM249 ^1^	HSV-1(F)BAC+EGFP	[125]
scFv to CEA @ gDΔ2-24	CEA	KNC ^1^	HSV-1(KOS)+gDY38C+gB:NT allele	[126]
scFv to EGFR @ gDΔ2-24	EGFR, EGFRvIII	KNE ^1^	HSV-1(KOS)+gDY38C+gB:NT allele	[126]
scFv to EGFR @ gDΔ6-38	EGFR	R-611 ^1^	HSV-1(F)BAC+EGFP	[127]
scFv to PSMA @gDΔ6-38	PSMA	R-593 ^1^	HSV-1(F)BAC+EGFP	[127]
scFv to EGFRvIII @ gDΔ6-38	EGFRvIII	R-613 ^1^	HSV-1(F)BAC+EGFP	[127]
**@ gD (double engineering)**				
scFv to HER2 and GCN4 peptide @ gD aa 24, or @ gDΔ35-39, or @ gDΔ214-223	HER2 and GCN4R	R-87 ^1^, R-89 ^1^, R-97 ^1^, R-99 ^1^	HSV-1(F)BAC+EGFP	[128]
**@ gH**				
scFv to HER2 @ gH aa 23	HER2	R-VG809 ^2^	HSV-1(F)BAC+mCherry+gDΔ6-38	[129]

GCN4 peptide @ gH aa 23	HER2 and GCN4R	R-213 ^2^	R-LM113	[130]
**@ gB**				
scFv to HER2 @ gB aa 43	HER2	R-909 ^2^	HSV-1(F)BAC+EGFP+gDΔ6-38	[131]

^1^ fully retargeted (retargeted to heterologous receptors, and detargeted from HSV natural receptors). ^2^ contains companion deletions in gD to achieve detargeting from HSV natural receptors. aa: amino acid residue. BAC: bacterial artificial chromosome. Δ: deletion.

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
