# Peer review of "Herpes Simplex Virus Oncolytic Immunovirotherapy: The Blossoming Branch of Multimodal Therapy"

_ijms, 2020, doi:10.3390/ijms21218310_

Round 1

Reviewer 1 Report

In this review the authors focus on what is known on oHSV-1 in immunovirotherapy. In general this is a nicely written review that touches on all the latest advances in the field. The paragraphs are well organized and easy to follow, the tables and figures are well made and help the reader to follow such a complex topic.

As an addition to the review, I suggest that the authors include a table (or add a column in the tables) in which oHSV-1s that have been or currently are in clinical trials are mentioned. In most cases they are mentioned in the text, but I think a tabular summary would make the data more accessible to readers.

I also suggest to specify throughout the text when a new oHSV-1 is introduced, e.g. by adding “oncolytic virus” or “oHSV” before the initialling. I also noticed that before the third paragraph all the names of oHSV-1 were mentioned in the text. However this is not the case in paragraph 3 (tropism retargeted, unattenuated, oHSVs). Is there a specific reason?

In addition, it is known that in order for HSV-1 vectors to be used as therapeutics, a higher purity must be achieved. Is it possible to include a few phrases about how these viruses are produced before going into clinical trials?

Now and then the references are written in the following way: (see […]). This is not a common style for citations and should be avoided. Moreover it is not used consistent throughout the text. Here are some lines noted: 29, 42, 45-46, 47, 88, 118, 140, 228, 239, 249, 363, 372, 393.

Line 33- 35: I found unprecise the statement “modifications in bacterial system” considering that CRISPR/Cas9-mediated editing of HSV-1 is performed in eukaryotic cells. I suggest to modify the sentence.

Line 36: This is the first time (excluding the abstract) in which the acronym oHSV-1 appears, but the full name is spelled in line 40. 

Line 46 – 48: The authors did not include the definition of the first generation according to some authors.

Line 75: The description of R7020 (NV1020) is not fully described in the table 1 compared to the text.

Line 132: The authors should mention that HCMV IRS1 (and TRS1) are functionally analogous to γ34.5.

Line 134: A space is missing for the 2.2 paragraph.

Reviewer 2 Report

General Comments

The manuscript is a thorough review on oncolytic HSV vectors and covers the field quite extensively. It is generally well written and relatively easy to follow despite the complicated nomenclature (not the authors’ fault!). The illustrations and tables make the reading easier.

However, the Conclusion section is really disappointing. It is very short and more like a “second Abstract” not presenting much of a view of where the field is going. There is no comparison to other viral vector systems. You would think that HSV is the only possible delivery system available after reading the review. Disappointingly, the approval of the first oncolytic immunotherapy is mentioned in a single (although long) sentence! Moreover, there are several points raised below, which need to be addressed in a revision of the manuscript.

The authors speak on P2, L56 of the deletion of UL23 as a “single mutation”, which is incorrect. The definition of mutation is “mistake in DNA copying” and does NOT include deletions (at least not large deletions, frame-shift mutations are another story).

I find it a bit odd that the description of the FDA approved HSV-based T-VEC is mentioned at the end of section 2.2 (P8), but in the section 2.3 dealing with T-VEC (P8-9) nothing is mentioned about T-VEC and the paragraph ends with a phase III trial. This is not chronologically very logical!

Specific Comments

P1, L14: “licensed” > “approved”

P1, L15: “tumors” > “cancer types”

P1, L24: “viruses and oncolytic viruses” > oncolytic viruses”

P1, L25: “it showed” > “it has shown”

P1, L27: “in virus” > “of virus”; “and in virus-host” > “and virus-host”

P1, L29: “, and more cryptic” > “. Additional cryptic”

P1, L31: “dispensable in cell culture” > “dispensable in expression vectors”

P1, L42: “recombinants” > “recombinant vectors”

P2, L44: “one or the other” > “one or another”

P2, L46: “others authors” > “other authors”

P2, L48: “recombinants” > “recombinant vectors”

P2, L51: Heading should be: “Conditionally replicating oHSVs with single or multiple mutations”

P2, L52: The sentence starting with “The first of.....” is too long and complicated. Please modify!

P2, L59: Define “PKR”

P2, L61: “It was soon” > “It became soon”; “recombinants’ genome” > “recombinant genome”

P2, L70: Sentence should read: “....more recently in young patients with extracranial solid tumors, showing no toxicity....”

P2, L77: “US2 to part” > “US2 as part”

P2, L87: “a HSV-1” > “an HSV-1”

P4, L93: “color” > “colored”

P4, L94: “products names” > “product names”

P4, L95: “example” > “examples”

P4, L96: Should read: “oHSVs with single and multiple mutations”

P5, L123: “placing γ134.5” > “placing the γ134.5”

P5, L134: The heading 2.2 should be moved to L135

P5, L137: The sentence starting with “This crucial safety....” does not make sense, please modify!

P5, L139: Move “with transgenes” after “mutations”

P7, L151: “of example armed” > “of examples of armed”

P7, L152: “proteins is” > “proteins are”

P8, L172: Remove comma before “too”

P8, L182: “as arming cytokine”?

P9, L220: I doubt that readers in general know what “sine qua non” means

P9, L220: “allow” > “will allow”

P10, L258: “contain” > “contains”

P12, L271: “infect cell” > “infect cells”; add comma after “however”

P12, L273: “with a similar” > “by similar”

P12, L278: “recombinants” > “recombinant vectors”

P12, L280: “recombinant” > “recombinant vector”

P12, L285: “lead” > “led”

P12, L287: “aa” > “amino acids”

P12, L296: Add comma after “addition”

P12, L297: “recombinants’ in vitro oncolytic activity” > in vitro oncolytic activity of recombinant vectors”

P12, L305: “under control” > “under the control”

P12, L312; “but expand” > “but can be expanded”

P13, L331: “intratumor” > “intratumoral”

P13, L334: “intratumor” > “intratumoral”

P13, L349: “a HSV-2” > “an HSV-2”

P14, L363; “as partner” > “as a partner”

P14, L368: Add comma after “addition”

P14, L380: “to chemotherapy and the oHSV” > “to chemotherapy or oHSV alone”

P14, L390: “antigens release” > “antigen release”

P15, L421: “advance” > “development”
